# Static and Eigenvalue Analysis of Electrostatically Coupled and Tunable Shallow Micro-Arches for Sensing-Based Applications

**DOI:** 10.3390/mi14050903

**Published:** 2023-04-23

**Authors:** Hassen M. Ouakad, Ayman M. Alneamy

**Affiliations:** 1Mediterranean Institute of Technology, South Mediterranean University, Tunis 1053, Tunisia; 2Department of Mechanical and Industrial Engineering, Sultan Qaboos University, Muscat P.O. Box 123, Oman; 3Department of Mechanical Engineering, Jazan University, Jazan 45142, Saudi Arabia; alneamy@jazanu.edu.sa

**Keywords:** MEMS, electrostatically-coupled resonator, bi-stable, mode veering, mode crossing

## Abstract

This paper investigated the mechanical performance of an electrostatically tunable microbeams-based resonators. The resonator was designed based on two initially-curved microbeams that are electrostatically coupled, offering the potential for improved performance compared to single-beam based resonators. Analytical models and simulation tools were developed to optimize the resonator design dimensions and to predict its performance, including its fundamental frequency and motional characteristics. The results show that the electrostatically-coupled resonator exhibits multiple nonlinear phenomena including mode veering and snap-through motion. A coexistence of two stable branches of solutions for a straight beam case was even obtained due to the direct effect of the coupling electrostatic force with the other curved beam. Indeed, the results are promising for the better performance of coupled resonators compared to single-beam resonators and offer a platform for future MEMS applications including mode-localized based micro-sensors.

## 1. Introduction

Micro-electromechanical systems (MEMS) resonators have emerged as a crucial component in a variety of technological applications, including wireless communication systems, biosensors and frequency resonators [1,2,3]. The earliest MEMS resonators were made of simple cantilever beams but, since then, many other types of MEMS resonators have been developed. These include various types of vibrating structures, such as clamped-clamped beams, ring resonators and torsional resonators [4,5]. One type of MEMS resonator that has garnered significant attention is the initially-curved microbeams resonator. Unlike straight beams, they can provide higher quality factors and lower motional resistances [6,7,8]. A curved beam can be obtained by buckling a straight beam under axial load (pre-stress) or via initial (stress-free) fabrication [9]. The existence of bistability in curved beams is a function of their dimensions and initial rise [10,11].

Analytical conditions for the existence of bistability in curved beams under electrostatic excitation and axial loads were derived by [12,13]. The performance of these micro-structures is affected by several design parameters including their mid-point initial rise, beam thickness, applied axial load, internal residual stress and excitation force. The designed dimensions and fabrication technique of the initially-curved microbeams have to be addressed carefully to ensure the existence of the snap-through motion.

These benefits have made them attractive for use in high-performance resonant structures making them more advantageous over their straight beam counterparts [14,15,16,17]. They have been extensively studied in recent years, investigating various aspects of their design, fabrication and performance. One important aspect of initially-curved microbeams resonators is their mechanical behavior. Indeed, these micro-structures can exhibit complex modes of vibration due to their initially-curved geometry, which can significantly impact their performance. Several studies have focused on developing analytical models and simulation tools to better understand the behavior of these resonators, including investigating the effects of curvature, length and thickness on their fundamental frequency and quality factor [18,19,20,21,22]. Coupled microbeams-based resonators have emerged as a promising type of MEMS resonator in recent years, offering several advantages over single beam resonators. In these resonators, two or more microbeams are coupled together, allowing for enhanced performance through the interaction between the beams. They have been used in a variety of applications, such as resonant sensors, filters and oscillators.

Using the concept of coupled resonators in MEMS sensors has attracted many researchers. For example, Ramezany et al. [23] introduced a proof-of-concept for a frequency modulated output MEMS sensor utilizing electrostatic coupling force between two clamped-clamped resonators. Li et al. [24] studied the exploitation of bifurcation jumping behavior in electrically-coupled microbeam resonators that overcomes the limitations of traditional methods based on frequency shift measurements to enhance the sensitivity. A skewed electrodes array has been utilized to improve the robustness of mass sensors in a parametrically-excited, mode-localized coupled resonator [25,26].

Karabalin et al. [27] presented an array of two nanoresonators that were coupled by a force and independently excited around their fundamental frequencies. They found that the linear and weakly nonlinear responses of a nanoresonator could be altered by the excitation of the other nanoresonators. Additionally, when two resonators were strongly excited in their nonlinear domain, the response curves became more complex. Baguet et al. [28] investigated the mass-sensing capability of an array of a few identical electrostatically-actuated microbeams in order to introduce a new class of MEMS-based coupled resonators. Rabenimanana et al. [29] studied the veering phenomena and the nonlinear dynamic response of mechanically-coupled MEMS resonators considering different actuation mechanisms.

Furthermore, several research papers have examined the possibility of mode localization in coupled MEMS structures [30,31]. Indeed, this phenomenon is defined as the confinement of vibration energy to one of the members of the coupled system in response to an external perturbation. This process can be accompanied by another closely related process called the eigenvalue curve veering [32]. This mode veering occurs when two linearly-coupled frequencies of the system approach each other and then deviate away as one of the system control parameter is varied. Recently, the veering phenomenon was examined as a main consequence of the mode localization phenomenon for coupled MEMS structures.

As a consequence, such intrinsic modes localization was correspondingly found to likely arise due to the presence of certain intrinsic nonlinearities within an array of coupled resonators [31,33]. As a result, a mass sensor based on the mode localization phenomenon was first proposed for two weakly mechanically-coupled cantilever beams [34]. Since then, different sensors assuming weakly-coupled MEMS structures have been investigated based on the mode localization process for mass sensors [35], force sensors [36], charge sensors [37] and accelerometers [38].

Overall, the research on coupled microbeams-based resonators has demonstrated their potential for use in a wide range of MEMS applications. Further advancements in their design and fabrication techniques could lead to improved performance and expanded applications for these resonators. Nevertheless, there is still a lack of comprehensive theoretical studies on mode localization based on coupled structures, especially when the coupling is due to the actuating electrostatic forcing. Consequently, the main objective of this paper is to investigate the design and examine the performance of MEMS electrically-coupled initially-curved microbeams resonators.

As shortcomings in the previous studies, tuning the mechanically coupled resonators and re-arranging the electrodes distribution required intensive optimization and higher actuation voltage. Therefore, the proposed coupled resonator design was based on two initially-curved microbeams that are electrostatically coupled through a single full electrode offering the potential for improved performance compared to single or multi-beams resonators. This study focused on the development of analytical models and simulation tools to optimize the coupled resonator parameters and to predict its performance, including its fundamental frequency and motional resistance. A reduced-order model was established and then eigenvalues of the two coupled systems were analyzed under different bias voltage values. The effect of the bias voltages and the initial curvatures of the coupled microbeams on the nonlinear behavior was thoroughly demonstrated.

Indeed, the performance of the initially-curved coupled resonator was analyzed with a focus on its mechanical properties, to demonstrate the feasibility of using it as MEMS sensors. The rest of the manuscript is comprised of the following sections. Section 2 introduces the coupled microbeam design, model, geometrical properties and operational mechanism. Section 3 presents the static and eigenvalue problem theoretical results along with a results discussion. Finally, Section 4 summarizes the work and suggests a few conclusions.

## 2. Device Geometrical Properties and Operational Mechanism

The coupled resonator consisted of two thin-beams both designed to be initially curved up (^+^ve configuration) or down (^−^ve configuration) and denoted as an upper and a lower beam, respectively. They are actuated by a single side-wall electrode as shown in Figure 1. The beams had a length of $\ell_{b}=1000$ µm and a width of b=30 µm. They both assumed a thickness of h=2 µm while their mid-point rises were varied from [−2.5:2] µm depending on an assumed initial outline. They were assumed to have a uniform cross-sectional area of *A* and an area moment of inertia *I*. The capacitor gap measured from the center line of each beam was set to $d_{l}=10$ µm, for the lower resonator and to $d_{u}=10$ µm, for the upper resonator. The device was assumed to be made of single crystal silicon with material properties and dimensions listed in Table 1.

The sensor operational mechanism assumed several nonlinear dynamic phenomena including snap-through, mode localization, hybridization and veering. This allowed the detection of any small perturbation in the vibration mode of the coupled resonators using different sensing scenarios. We noted that the selectivity and sensitivity are totally dependent on the coupling term between the resonators’ geometrical parameters, mechanical and material properties and the electrostatic coupling force. To operate the proposed design as mass sensors, we expected that any small perturbation in the target added mass would lead to veering, crossover and/or snap-through phenomena and, therefore, a change in the vibration mode.

The proposed design can also be operated depending on the vibrating mode of the microbeams including symmetric and anti-symmetric modes, respectively. Assuming a small perturbation to the total mass on one of the two beams and tuning the static component of the actuated force reduces the fundamental frequency. As a result, the coupled resonators vibrates at the lowest vibration mode following the weaker energy channel. This causes an interesting phenomenon well-known as mode-localization. In fact, this process confines the system’s fundamental frequency on the first resonator for the first mode and on the second one for the second mode; in turn, enhancing the electrostatic coupling term between the two beams and optimizing their geometric dimensions leading to high-class sensitive MEMS devices.

### 2.1. Mathematical Modeling

Following the procedure developed by [39,40,41] and Newton’s second law associated with the Euler Bernoulli’s beam theory, we wrote the equations of the motion describing the transverse deflections in the absence of the electrostatic fringing filed of the lower micro-resonator as follows:(1)$$\begin{array}{cc} \rho A{\hat{\ddot{w}}}_{l}+\hat{c}{\hat{\dot{w}}}_{l}+EI{\hat{w}}_{l}^{⁗}= & \frac{EA}{2\ell_{b}}\left[{\hat{w}}_{l}^{\prime \prime }\int_{0}^{\ell_{b}}\left({\hat{w}}_{l}^{\prime 2}-2{\hat{w}}_{l}^{\prime }\right)\hat{x}\right]+\frac{\varepsilon bV_{l}^{2}}{2{\left(d_{l}-{\hat{w}}_{l}\pm {\hat{w}}_{l\circ }\right)}^{2}}\\ & -\frac{\varepsilon bV_{u}^{2}}{2{\left(d_{u}+{\hat{w}}_{l}\pm {\hat{w}}_{l\circ }-{\hat{w}}_{u}\pm {\hat{w}}_{u\circ }\right)}^{2}} \end{array}$$
where the two beams are assumed to be uniform and isotropic. The initial shape of the lower-beam was laid out to follow the expression
$${\hat{w}}_{l\circ }=\frac{b_{l\circ }}{2d_{l}}\left[1-\cos\left(2\pi \hat{x}\right)\right]$$
while the upper-beam was laid out to follow the expression
$${\hat{w}}_{u\circ }=\frac{b_{u\circ }}{2d_{u}}\left[1-\cos\left(2\pi \hat{x}\right)\right]$$
and the associated boundary conditions at the two supports are
(2)$${\hat{w}}_{l}\left(0,\hat{t}\right)=0\;,\;{\hat{w}}_{l}^{\prime }\left(0,\hat{t}\right)=0\;,\;{\hat{w}}_{l}\left(\ell_{b},\hat{t}\right)=0\;,\;{\hat{w}}_{l}^{\prime }\left(\ell_{b},\hat{t}\right)=0$$

The equations of motion describing the transverse deflections of the upper micro-resonator can be also written as
(3)$$\begin{array}{cc} \rho A{\hat{\ddot{w}}}_{u}+\hat{c}{\hat{\dot{w}}}_{u}+EI{\hat{w}}_{u}^{⁗}= & \frac{EA}{2\ell_{b}}\left[\left({\hat{w}}_{u}^{\prime \prime }-{\hat{w}}_{u\circ }^{\prime \prime }\right)\int_{0}^{\ell_{b}}\left({\hat{w}}_{u}^{\prime 2}-2{\hat{w}}_{u\circ }^{\prime }{\hat{w}}_{u}^{\prime }\right)d\hat{x}\right]\\ & +\frac{\varepsilon bV_{u}^{2}}{2{\left(d_{u}+{\hat{w}}_{l}\pm {\hat{w}}_{l\circ }-{\hat{w}}_{u}\pm {\hat{w}}_{u\circ }\right)}^{2}} \end{array}$$
with associated boundary conditions listed as
(4)$${\hat{w}}_{u}\left(0,\hat{t}\right)=0\;,\;{\hat{w}}_{u}^{\prime }\left(0,\hat{t}\right)=0\;,\;{\hat{w}}_{u}\left(\ell_{b},\hat{t}\right)=0\;,\;{\hat{w}}_{u}^{\prime }\left(\ell_{b},\hat{t}\right)=0,$$
where *A* is the beam cross-sectional area, ρ denotes the mass density and *I* represents the moment of inertia, which is equal to $\frac{bh^{3}}{12}$. Equation (Equation 1) shows that the lower resonator is electrostatically coupled with the upper resonator through the forcing component $V_{u}$. Indeed, it confirms that, as the lower beam is electrostatically excited, the upper beam will respond to it resulting in rich static and dynamic behavior. We noted that the strength of the coupling force is dependent on the excitation level applied by the side-wall electrode. On the other hand, changing the value of the electrostatic force coupling term could lead to several excitation scenarios of the coupled resonator as will be discussed in the following sections. Note that the (±) sign appears in Equations (Equation 1) and (Equation 3), referring to the initial curvatures of the coupled resonator as shown in Figure 1.

### 2.2. Normalization Equations of Motions

Next, we normalized the transverse equations of motion governing the coupled resonator response, Equations (Equation 1) and (Equation 3), to reduce the numerical error and to accelerate the computational time [42]. For convenience, we introduced the following nondimensional variables:(5)$$w_{l}=\frac{{\hat{w}}_{l}}{d_{l}}\;,\;w_{u}=\frac{{\hat{w}}_{u}}{d_{u}}\;,\;x=\frac{\hat{x}}{\ell_{b}}\;,\;t=\frac{\hat{t}}{T}$$
where $T=\sqrt{\rho bh\ell_{b}^{4}/EI}$ and it is a time scale parameter. Substituting Equation (Equation 5) into Equations (Equation 1)–(Equation 4) yields to the nondimensional equation of motion that governs the transverse deflection of the lower resonator, which can be written as:(6)$$\begin{array}{cc} {\ddot{w}}_{l}+c{\dot{w}}_{l}+EIw_{l}^{⁗}= & \alpha_{1}\left[w_{l}^{\prime \prime }\int_{0}^{1}\left(w_{l}^{\prime 2}-2w_{l}^{\prime }\right)dx\right]+\frac{\alpha_{2}V_{l}^{2}}{{\left(1-w_{l}\pm w_{l\circ }\right)}^{2}}\\ & -\frac{\alpha_{2}V_{u}^{2}}{{\left(\frac{d_{u}}{d_{l}}+w_{l}\pm w_{l\circ }-w_{u}\pm w_{u\circ }\right)}^{2}} \end{array}$$
and its associated boundary conditions are, respectively,
(7)$$w_{l}\left(0,t\right)=0\;,\;w_{l}^{\prime }\left(0,t\right)=0\;,\;w_{l}\left(1,t\right)=0\;,\;w_{l}^{\prime }\left(1,t\right)=0$$
and nondimensional equation of motion that governs the transverse deflection of the upper resonator is
(8)$$\begin{array}{cc} {\ddot{w}}_{u}+c{\dot{w}}_{u}+EIw_{u}^{⁗}= & \alpha_{1}\left(w_{u}^{\prime \prime }-w_{u\circ }^{\prime \prime }\right)\left[\int_{0}^{1}\left(w_{u}^{\prime 2}-2w_{u\circ }^{\prime }w_{u}^{\prime }\right)dx\right]\\ & +\frac{\alpha_{2}V_{u}^{2}}{{\left(\frac{d_{u}}{d_{l}}+w_{l}\pm w_{l\circ }-w_{u}\pm w_{u\circ }\right)}^{2}} \end{array}$$
with associated boundary conditions listed as
(9)$$w_{u}\left(0,t\right)=0\;,\;w_{u}^{\prime }\left(0,t\right)=0\;,\;w_{u}\left(1,t\right)=0\;,\;w_{u}^{\prime }\left(1,t\right)=0$$
where the nondimensional coefficients appear in Equations (Equation 6) and (Equation 8), and are defined in Table 2.

### 2.3. Reduced-Order Model

A reduced-order model (ROM) based on a Galerkin method is utilized to solve the nondimensional equations of motion, (Equation 6) and (Equation 8). This technique discretizes the equation of motion in terms of a finite number of degrees-of-freedom describing the amplitude of mode shapes that satisfy the boundary conditions. In this case, we chose straight beam mode shapes $\phi_{i}\left(x\right)$ as basis functions in a Galerkin expansion.

Then, we solve for the mid-point static deflection of the coupled resonator as a function of the static voltages ldc and udc. This is carried out by eliminating the time derivatives from the equations of motion, Equations (Equation 6) and (Equation 8). It results in a static equation for the lower resonator, which is electrostatically coupled with the upper resonator through udc voltage as follows:(10)$$\begin{array}{cc} w_{ls}^{⁗}=\alpha_{1}\left[w_{ls}^{\prime \prime }\int_{0}^{1}\left(w_{ls}^{\prime \prime }-2w_{ls}^{\prime }\right)dx\right] & +\frac{\alpha_{2}ldc^{2}}{{\left(1-w_{ls}\pm w_{l\circ }\right)}^{2}}-\frac{\alpha_{2}udc^{2}}{{\left(\frac{d_{u}}{d_{l}}+w_{ls}\pm w_{l\circ }-w_{us}\pm w_{u\circ }\right)}^{2}} \end{array}$$
and it is subjected to the following boundary conditions:(11)$$w_{ls}\left(0,t\right)=0\;,\;w_{ls}^{\prime }\left(0,t\right)=0\;,\;w_{ls}\left(1,t\right)=0\;,\;w_{ls}^{\prime }\left(1,t\right)=0$$

Similarly, the equation describing the static equilibria of the upper beam under the effect of the upper static voltage can be written as:(12)$$\begin{array}{c} w_{us}^{⁗}=\alpha_{1}\left(w_{us}^{\prime \prime }-w_{u\circ }^{\prime \prime }\right)\left[\int_{0}^{1}\left(w_{us}^{\prime \prime }-2w_{u\circ }^{\prime }w_{us}^{\prime }\right)dx\right]+\frac{\alpha_{2}udc^{2}}{{\left(\frac{d_{u}}{d_{l}}+w_{ls}\pm w_{l\circ }-w_{us}\pm w_{u\circ }\right)}^{2}} \end{array}$$
and it is subjected to the following boundary conditions:(13)$$w_{us}\left(0,t\right)=0\;,\;w_{us}^{\prime }\left(0,t\right)=0\;,\;w_{us}\left(1,t\right)=0\;,\;w_{us}^{\prime }\left(1,t\right)=0$$

Then, we descretize the static deflections of the two microbeams in terms of the Galerkin expansion as:(14)$$\begin{array}{cc} w_{ls} & =\sum_{i=1}^{N}\phi_{i}\left(x\right)q_{li}\;;\;i=1,\ldots ,N\\ w_{us} & =\sum_{i=1}^{N}\phi_{i}\left(x\right)q_{ui}\;;\;i=1,\ldots ,N \end{array}$$
where $q_{li}$ are modal coordinates for the lower beam and $q_{ui}$ are modal coordinates for the upper beam. Substituting these transformation forms into Equations (Equation 10)–(Equation 13), multiplying both sides of Equation (Equation 10) by $\left[{\left(1-w_{ls}\pm w_{l\circ }\right)}^{2}\times {\left(\frac{d_{u}}{d_{l}}+w_{ls}\pm w_{l\circ }-w_{us}\pm w_{u\circ }\right)}^{2}\right]$ and multiplying both sides of Equation (Equation 12) by ${\left(\frac{d_{u}}{d_{l}}+w_{ls}\pm w_{l\circ }-w_{us}\pm w_{u\circ }\right)}^{2}$ avoids numerical errors in the response near the singularity. Then, multiplying the resulting equations by the mode shapes $\phi_{j}$ and carrying out the integration over the beam span results in *N* algebraic equations describing the equilibrium position for the lower beam as:(15)$$\begin{array}{cc} \; & \int_{0}^{1}\phi_{j}[\left({\left(1-\sum_{i=1}^{N}\phi_{i}q_{li}\pm w_{l\circ }\right)}^{2}{\left(\frac{d_{u}}{d_{l}}+\sum_{i=1}^{N}\phi_{i}q_{li}\pm w_{l\circ }-\sum_{i=1}^{N}\phi_{i}q_{ui}\pm w_{u\circ }\right)}^{2}\right)\;(\sum_{i=1}^{N}\phi_{i}^{iv}q_{li}\\ & -\alpha_{1}\left(\sum_{i=1}^{N}\phi_{i}^{{}^{\prime \prime }}q_{li}\right)\int_{0}^{1}\left({\left(\sum_{i=1}^{N}\phi_{i}^{{}^{\prime }}q_{li}\right)}^{2}-2\sum_{i=1}^{N}\phi_{i}^{{}^{\prime }}q_{li}\right)dx)+\frac{\alpha_{2}ldc^{2}}{{\left(1-\sum_{i=1}^{N}\phi_{i}^{{}^{\prime \prime }}q_{li}\pm w_{l\circ }\right)}^{2}}\\ & -\frac{\alpha_{2}udc^{2}}{{\left(\frac{d_{u}}{d_{l}}+\sum_{i=1}^{N}\phi_{i}q_{li}\pm w_{l\circ }-\sum_{i=1}^{N}\phi_{i}q_{ui}\pm w_{u\circ }\right)}^{2}}]=0 \end{array}$$
and for the upper beam as
(16)$$\begin{array}{cc} \; & \int_{0}^{1}\phi_{j}[{\left(\frac{d_{u}}{d_{l}}+\sum_{i=1}^{N}\phi_{i}q_{li}\pm w_{l\circ }-\sum_{i=1}^{N}\phi_{i}q_{ui}\pm w_{u\circ }\right)}^{2}\;(\sum_{i=1}^{N}\phi_{i}^{iv}q_{ui}-\alpha_{1}\left(\sum_{i=1}^{N}\phi_{i}^{{}^{\prime \prime }}q_{ui}-w_{u\circ }^{\prime \prime }\right)\\ & \int_{0}^{1}\left({\left(\sum_{i=1}^{N}\phi_{i}^{{}^{\prime }}q_{ui}\right)}^{2}-2w_{u\circ }^{\prime }\sum_{i=1}^{N}\phi_{i}^{{}^{\prime }}q_{ui}\right)dx)+\frac{\alpha_{2}udc^{2}}{{\left(\frac{d_{u}}{d_{l}}+\sum_{i=1}^{N}\phi_{i}q_{li}\pm w_{l\circ }-\sum_{i=1}^{N}\phi_{i}q_{ui}\pm w_{u\circ }\right)}^{2}}]=0 \end{array}$$
These equations are then solved for $q_{li}$ and $q_{ui}$ as functions of the static voltage to obtain the static deflections of the lower and upper resonators. We acquire the coupled resonator small vibrations problem by resolving its deflection into a static component $w_{s}\left(x\right)$ and a dynamic component $w_{d}\left(x,t\right)$ for the two beams as follows:(17)$$\begin{array}{cc} w_{l} & =w_{ls}\left(x\right)+w_{ld}\left(x,t\right)\\ w_{u} & =w_{us}\left(x\right)+w_{ud}\left(x,t\right) \end{array}$$

Substituting Equation (Equation 17) into Equations (Equation 15) and (Equation 16), dropping the damping coefficients, expanding the right hand-side term around the dynamic component using a Taylor series, dropping the high order terms and retaining only up to the linear term in $w_{ld}$ and $w_{ud}$ to obtain the forced eigenvalue problem. Here, the two microbeams oscillate around their respective static equilibria. Then, a similar procedure to that used in the static analysis can be carried out to develop the reduced-order model of the eigenvalue problem.

## 3. Results and Discussions

In this section, we analytically investigated the static deflections and the eigenvalues of the coupled resonator under static voltage waveforms for the three proposed scenarios discussed in Section 2. The actuation signal varies on the upper resonator while the lower resonator remains un-actuated or biased with 40 (V). For this particular case study, our main objective was to understand how the coupled resonator behaves and interacts statically as its initial curvatures and actuation levels are varied. In addition, we tracked the change in their fundamental frequencies that could possibly lead to nonlinear phenomena including snap-through, veering, mode localization or mode crossing as the control parameters vary across the two microbeams. This provides insight into the usability of such a design as a mass sensor.

### 3.1. Static Analysis

The static deflections of the lower and upper resonators mid-points $w_{ls}\left(0.5\right)$ and $w_{us}\left(0.5\right)$ excited by a distributed electrostatic force were computed through simultaneously solving the coupled Equations (Equation 15) and (Equation 16) using three symmetric modes ($\phi_{1}$, $\phi_{3}$ and $\phi_{5}$) and one anti-symmetric ($\phi_{2}$) mode in the Galerkin’s expansion. According to Younis et al. [42], at least three symmetric modes are required for satisfactory model convergence. Using two modes in the model results in quantitative errors, especially when the beam undergoes large deflection. In the below, we considered five different configurations in terms of initial curvatures, curved-up or down, varied the excitation signal along the upper beam and maintained a constant voltage for the lower beam.

Table 3 summarizes the mid-point rise of each beam under study. This was a mandatory step towards initiating a general design procedure to ensure promising static and dynamic behavior for the electrostatically-coupled MEMS resonator and to determine the minimum threshold for the initial curvature and other geometric parameters. These provide details about the existing of mode veering and snap-through motion as well the effect of the mid-point rise.

Assuming a straight lower beam and 1 (µm) mid-point rise for the upper beam leads to a single stable equilibrium for both beams as the upper static voltage varies and the lower static voltage remains un-actuated, shown with green lines (—) and, when biased with 40 V, displayed with magenta lines (—), respectively, in Figure 2a. The figure shows that the upper beam deflection increases in the (${}^{+}\mathrm{ve}$) positive direction and moves away from the center line as the voltage increases and vice versa for the lower beam deflection.

We observed one stable branch of solution for each beam as it terminates and loses its stability through a saddle-node bifurcation. At this point, the stable branch meets the unstable branch of solutions and, therefore, the beams come into contact with the side-wall electrode. We note that this level of mid-point rise is not sufficient to activate snap-through motion. To further investigate the effect of the initial rise, we considered a curved-up upper beam with 2 µm mid-point rise while maintaining a straight beam configuration for the lower beam, which represents design 2 as listed in Table 3. This, in fact, results in two stable branches of solutions as the upper static voltage udc increases and is illustrated in Figure 2b. It happens that even the lower beam remains in the straight position and is un-actuated or biased with 40 (V).

The reason for the existence of the bi-stability is that the upper beam has a larger length than the actual length between the two supports. In addition, due to the electrostatic force coupling term appearing in Equation (Equation 10), the upper microbeam has a direct effect on the static response of the lower beam through udc. This also leads to the coexistence of two stable branches of solutions for the lower beam, shown as a green line (—) for the un-actuated case and as a magenta line (—) for the case of 40 (V).

Figure 2b also shows two stable and unstable branches of solutions before and after the snap-through motion. We note that varying the upper beam voltage while maintaining a constant static voltage along the lower beam shows that the mid-point deflection of the lower beam ($w_{ls}$) decreases, (${}^{-}\mathrm{ve}$) negative direction, as the upper voltage increases along the first branch of stable equilibria, corresponding to the beam initial curvature configuration. Additionally, the upper beam deflection increases until the udc voltage reaches 42.95 (V), where the two microbeams jump to a second equilibrium commensurate to the initial counter-curvature through a saddle-node bifurcation.

At this point, the stable branch meets the first unstable branch of the solution. This jump is a basic characteristic of the snap-through mechanism. Further increasing in the actuation voltage leads to more deflection along the second stable branch until it terminates in a second saddle-node bifurcation bounding as the pull-in instability. Beyond this point, there is no physical existing solution because the two beams collapse. The figure also shows that the two beams start deflecting from a zero position for the un-actuated lower beam.

It is worth noting that decreasing the actuation upper voltage after jumping to the second stable configuration results in a second snap-through bifurcation named snap-back. However, it occurs at a lower voltage corresponding to 41.41 (V) compared to the snap-through threshold. The resulting gap between these two points is characterized as a narrow hysteresis region, which may not be suitable for MEMS based sensors. On the other hand, a similar behavior is observed when the lower static voltage ldc sets to 40 (V). However, the lower beam starts deflecting from a new position corresponding to 1.8 (µm) and requires more voltage to reach the snap-through point. This is expected because the dc voltage changes the beam’s curvature and as a result it becomes stiffer.

Setting the initial curvature of the two beams equal to 2 (µm) results in a similar static behavior to that of design 2, as shown in Figure 3a. However, the snap-through, snap-back and pull-in points occur at lower voltages. We note that the static stable deflection of the lower beam is small before and after the snap-through, compared to that of the upper beam. This is true for the lower beam that remains un-actuated or biased with 40 (V) because it is curved towards the side-wall electrode. In fact, this arrangement requires a lower voltage to pull it down and to reach the bifurcation points. For these dimensions, we found two stable branches of solutions indicating that the selected parameters are sufficient enough to activate the snap-through motion.

Alternatively, the lower beam deflects more than the upper beam when it curves down, while the upper beam is set to the straight configuration. This results in a gap measured from the lower beam mid-point to the side-wall electrode larger than that of design 3. Setting the lower static voltage to zero, the static deflections of the two beams move away from the center line and in opposite directions as clearly shown in Figure 3b and marked as green lines (—). We also note that the design under these actuation conditions has a snap-through behavior with two stable branches of solutions.

However, the snap-through mechanism does not exist anymore as the lower beam static voltage is set to 40 (V). It means that the two beams move directly to the pull-in voltage before they snap toward the second equilibria and are marked as magenta lines (—) in Figure 3b. On the other hand, the second stable branches of solutions can be reached if the two beams start as deflected from a pre-defined counter-curvature position.

Curving down the lower beam mid-point rise to $b_{l\circ }=-2.5\;$(µm) while maintaining the initial mid-point rise of the upper beam equal to $b_{u\circ }=2$ (µm) results in a complex static response as shown in Figure 3c. This confirms that, for these kinds of beam dimensions, the snap-through motion is not reachable specifically when the lower beam is biased with 40 (V), marked with magenta lines (—) for the two beams’ deflections. Indeed, to activate the snap-through motion statically, the ratio between the beam mid-point rise, length, width and thickness has to be within a certain value. This value should guarantee a dual energy well, which results in a bi-stable structure-based structure.

### 3.2. Fundamental Frequencies under Electrostatic Force

Subsequently, we investigated the impact of the static voltage on the fundamental frequencies of the coupled resonator designed with a different mid-point rise similar to that studied in the static part. Towards this, we substituted the static results obtained by employing an ROM with three symmetric and one anti-symmetric mode into the coupled linear eigenvalue problem and then solved for the corresponding eigenvalues.

The first three fundamental frequencies ($f_{i}$) were calculated using the ROM and they correspond to the first and second in-plane symmetric modes and the first in-plane anti-symmetric mode, respectively. Two actuation scenarios were utilized. The first one considered the case when the voltage was directly applied to the upper beam while the lower beam remained un-actuated and the second case was similar to that of the first case; however, the lower beam was biased with 40 (V).

Considering the first actuation case and the initial parameters of design 1 in Table 3, the first symmetric fundamental frequency, $f_{1}=17.50$ (kHz) at udc=0 (V), of the lower beam is found to decrease simultaneously as the upper voltage udc is increased. We note that the lower beam frequency goes to zero at the pull-in voltage when the lower beam voltage sets to zero, as outlined with a solid blue line (—). This is mainly because of the initial rise of the upper beam mid-point is not high enough to activate snap-through motion.

On the other hand, the upper beam fundamental frequency, $f_{1}=20.37$ (kHz) at udc=0 (V), decreases until it reaches a zone when its value is closed to that of the lower beam indicating a veering phenomenon. Then, it increases drastically as the upper static voltage increases and is marked as a solid orange line (—) in Figure 4a. A manifestation of such a frequency approach is known in the literature as the frequency veering phenomenon. This is a classical behavior of the veering phenomenon where two modes are approaching each other and then veer away as the control parameters change. We found that a frequency gap of 1.05 (kHz) occurs at udc=19.7(V).

Replicating the previous analysis with a 40 (V)-biased lower beam results in a similar behavior without any indication of the veering phenomenon. An exact behavior has been found for the second ($f_{2}$) and third ($f_{3}$) eigenvalues, respectively. However, the veering zone starts to develop at a higher voltage for each frequency compared to that of the first in-plane symmetric mode. The results corresponding to this case study are marked as symbols in Figure 4a. It is also worth noting that the higher eigenvalues do not approach zero because the geometric nonlinearities overcomes the electrostatic force nonlinearities and the coupled resonator has only one stable equilibrium considering these dimensions.

On the other hand, increasing the upper beam mid-point rise to 2 (µm) results in an interesting nonlinear behavior characterized by snap-through motion as shown in Figure 4b for the two beams. While the lower beam remains un-actuated, we found that the fundamental frequency continuously drops along the first stable equilibria as the upper voltage increases until it reaches zero at the snap-through threshold, marked as a solid blue line (—). Then, it increases along the second stable branch of equilibria when the upper beam jumps and changes its initial configuration to the counter-curvature and then decreases as the voltage approaches the pull-in voltage.

Decreasing the voltage below the snap-through point results in a further reduction in the fundamental frequency until it reaches zero and then regains stability when jumping-back to its initial configuration at a point called snap-back. Between these two thresholds, a hysteresis region is developed. In addition, a similar behavior is observed for the upper resonator denoted by a solid orange line (—). However, its fundamental frequency is higher than that of the first design at the zero voltage due to its initial imperfection.

Figure 4b illustrates that the higher frequencies developed an exact variation trends compared to that of the first mode. We note that due to the electrostatic coupling term, the lower beam has two stable branches of solutions even it is initially designed as a straight beam. We also found that the veering occurs only between the symmetric modes of the two beams as they approach each other and veer away as the control parameters changes. However, this is not the case for the anti-symmetric mode as the two branches move away from each other regardless to the value of the control parameter, which in this case is the upper voltage (udc).

Additionally, in the neighborhood of the veering zone, we have found that the minimum frequency gap between the two frequencies of the first symmetric mode is approximately Δf=2.1kHz and occurs at a voltage of udc=32.7(V) and for the second symmetric mode is approximately Δf=0.81kHz and occurs at a voltage of udc=24.64(V), respectively. Considering the second actuation case leads to a similar behavior; however, the lower beam fundamental frequencies are higher than those when it is un-actuated. This is expected because the beam is biased in this case and it is stiffer compared to the first case.

Curving up the mid-points of the two beams with 2 (µm) leads to a complex situation. For example, when the lower beam is un-actuated and the upper beam voltage is set to zero, the fundamental frequencies start from the same value, corresponding to 26.89 (kHz). This confirms that the design under these assumptions is symmetric. It is clearly shown in Figure 5a that varying the upper voltage shows that the frequency corresponding to the upper beam decreases until it reaches zero at the snap-through and is marked as orange (—) in the figure. We note that the frequency of the lower beam increases even if its voltage is set to zero. No insight about the veering nor crossing behavior has been observed.

However, biasing the lower beam with 40 (V) changes the behavior of the coupled resonator. We found that the lower fundamental frequency at zero upper voltage is less than that when it is un-actuated. This is expected because the beam is closer to the side-wall electrode and it becomes easy to pull it down. Then, increasing the upper static voltage leads to a further reduction in the upper beam fundamental frequency and an increase in the lower fundamental frequency.

Tracking these two branches of solutions results in a veering phenomenon as the two frequencies approach each other. Then, they move away as the control parameter increases and reaches the snap-through point. We note that this scenario is repeatable for the higher frequencies as shown in Figure 5a. Indeed, the design with these dimensions is a suitable candidate to activate the snap-through, which can be used to design a high sensitive coupled resonator.

On the other hand, a noticeable change in the stability comportment is observed for a design with a −2 (µm) lower beam mid-point rise and a straight upper beam. These dimensions correspond to design 4. We found that the resonator developed a snap-through behavior when the lower beam is un-actuated as shown in Figure 5b. This is not the case when the lower beam is biased. The system here goes directly to the pull-in region before it snaps towards the second equilibrium. This indicates that the snap-through region is not directly accessible. However, to activate the bi-stability mechanism, the resonator can be operated from a pre-deflected shape corresponding to the second equilibrium.

Alternatively, curving up the upper beam and curving down the lower beam with two different initial rises results only in a single equilibrium. This confirms that the device with this arrangement, design 5, is not suitable and sufficient to activate any snap-through motion.

## 4. Conclusions

This work investigated the structural nonlinear behavior of electrostatically-coupled initially curved micro-beams. The static and eigenvalue problems were numerically computed and solved. The results showed that, with proper tuning of the coupling DC loads and the respective arches’ initial curvatures, one can alter the overall structural stability of the resonator from having a one-stable state to two-stable states with snap-through/snap-back hysteretic capabilities. Variation of the eigenfrequencies also showed the potential of obtaining frequency veering states potentially of interest for the design of mode-localized-based MEMS sensors/harvesters. The proposed electrically-coupled and geometrically tunable design has potential characteristics that could fit several MEMS applications ranging from mode-localized mass/gas sensors, micro energy harvesters, and dual-state micro-resonators. Future work will include the examination of the mode-localization capabilities of the design through carrying out a dynamic analysis to further examine its suitability for MEMS sensing applications.

## Figures and Tables

**Figure 1 micromachines-14-00903-f001:**
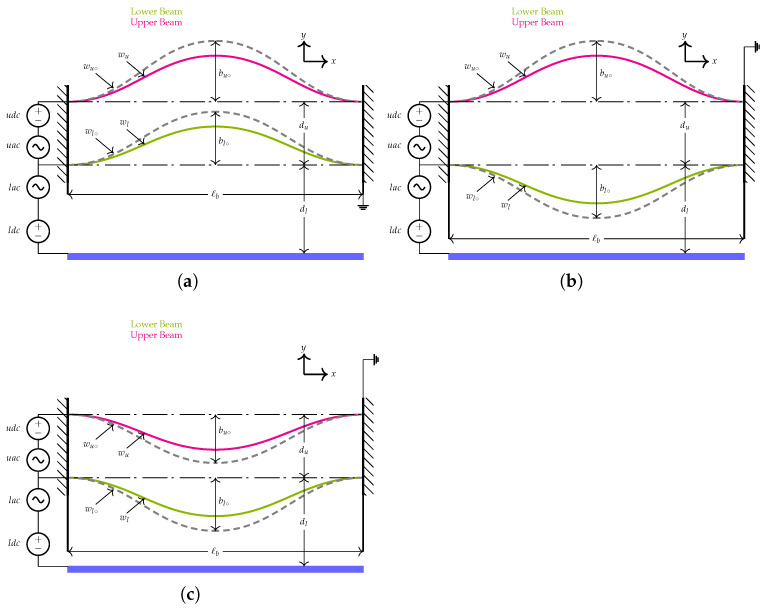
Schematics showing the proposed sensor and the electrical connection for (**a**) two resonators curved-up (^+^ve configuration) representing the first design, (**b**) upper resonator curved-up (^+^ve configuration) and the lower resonator curved-down (^−^ve configuration) representing the second design and (**c**) both resonators curved-down (^−^ve configuration) representing the third design.

**Figure 2 micromachines-14-00903-f002:**
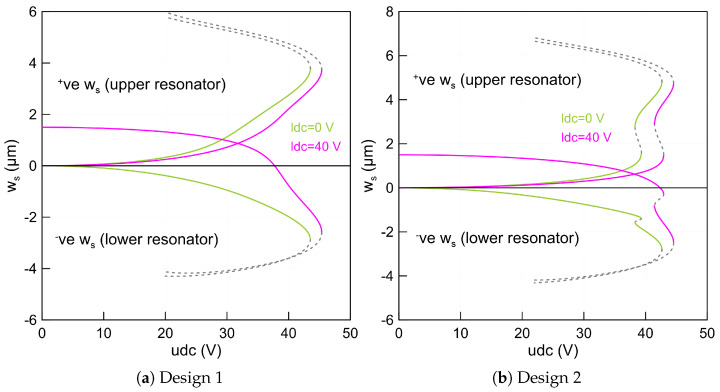
The beam mid-point deflections of the lower beam $w_{ls}\left(0.5\right)$ and of the upper beam $w_{us}\left(0.5\right)$ as a function of the upper static voltage (udc) using three-symmetric and one anti-symmetric modes in the ROM and a lower voltage set to zero marked as green lines (—) and 40 (V) marked as magenta lines (—) for: (**a**) straight lower beam and 1 (µm) upper beam mid-point rise and (**b**) straight lower beam and 2 (µm) upper beam mid-point rise. The stable static equilibria of the two beams are marked with solid lines while the unstable static equilibria are marked with dashed gray lines (- - -).

**Figure 3 micromachines-14-00903-f003:**
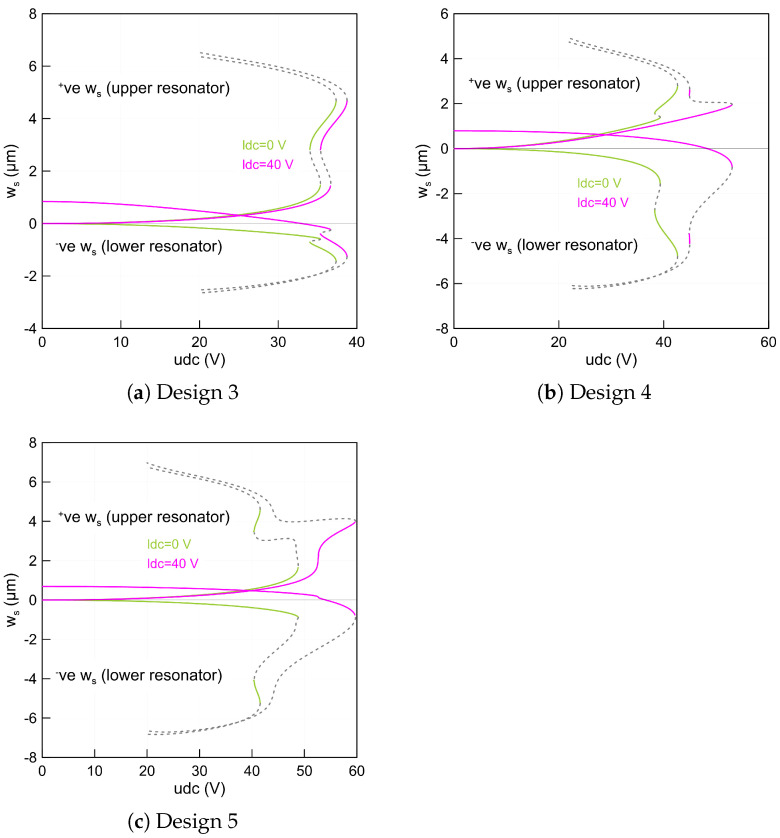
The beam mid-point deflections of the lower beam $w_{ls}\left(0.5\right)$ and of the upper beam $w_{us}\left(0.5\right)$ as a function of the upper static voltage using three-symmetric and one anti-symmetric mode in the ROM and a lower voltage set to zero marked as green lines (—) and 40 (V) marked as magenta lines (—) for: (**a**) design 3 with 2 (µm) upper and lower beams mid-points rise, (**b**) design 4 with straight upper beam and −2 (µm) lower beam mid-point rise and (**c**) design 5 with −2.5 (µm) lower and 2 (µm) upper beams mid-points rise. The stable static equilibria of the two beams are marked with solid lines while the unstable static equilibria are marked with dashed gray lines (- - -).

**Figure 4 micromachines-14-00903-f004:**
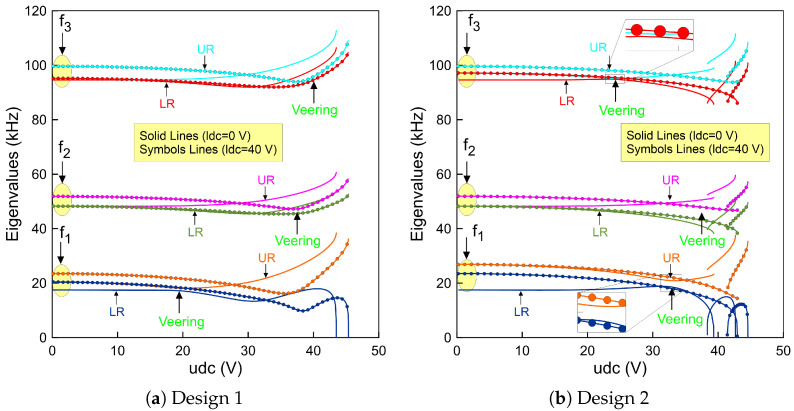
The variation of the fundamental frequencies ($f_{1},\;f_{2}$ and $f_{3}$) of the lower beam and the upper beam as a function of the upper static voltage using three-symmetric and one anti-symmetric modes in the ROM and a lower voltage set to zero for: (**a**) design 1 made of a straight lower beam and 1 (µm) upper beam mid-point rise and (**b**) design 2 made of a straight lower beam and 2 (µm) upper beam mid-point rise.

**Figure 5 micromachines-14-00903-f005:**
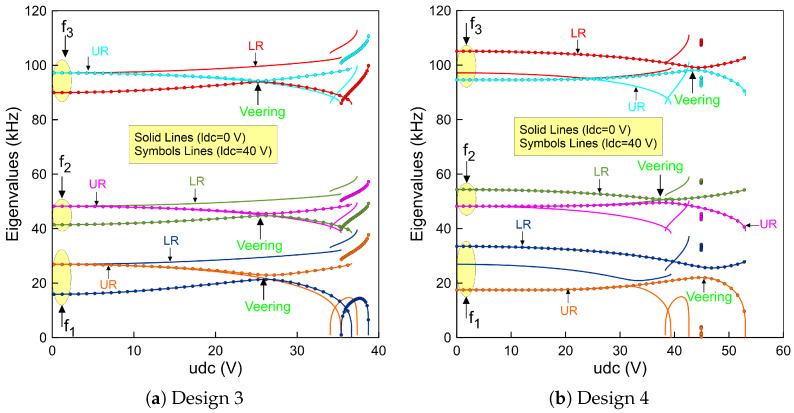
The variation of the fundamental frequencies ($f_{1},\;f_{2}$ and $f_{3}$) of the lower beam and the upper beam as a function of the upper static voltage using three-symmetric and one anti-symmetric mode in the ROM and a lower voltage set to zero for: (**a**) design 3 made of 2 (µm) lower and upper beams mid-points rise and (**b**) design 4 made of a straight upper beam and a −2 (µm) lower beam mid-point rise.

**Table 1 micromachines-14-00903-t001:** Coupled resonator material properties and dimensions.

Description	Value
Density (ρ)	2330 kg/m^3^
Young’s Modulus (*E*)	129 GPa
Dielectric constant of the air (ε)	8.854 $\times \;10^{-12}$ F/m
Resonators’ length ($\ell_{b}$)	1000 µm
Resonators’ width (*b*)	30 µm
Resonators’ thickness (*h*)	2 µm
Capacitor gap ($d_{u}$ and $d_{l}$)	10 µm
Upper resonator mid-point rise ($b_{u\circ }$)	−2:2 µm
Lower resonator mid-point rise ($b_{l\circ }$)	−2.5:2 µm

**Table 2 micromachines-14-00903-t002:** Nondimensional coefficients appear in the equations of motion.

Description	Expression
Viscous damping coefficient (*c*)	$c=\frac{\hat{c}\ell_{b}^{4}}{EIT}$
Mid-plane stretching coefficient ($\alpha_{1}$)	$\alpha_{1}=6{\left(\frac{d_{l}}{h}\right)}^{2}$
Electrostatic force coefficient ($\alpha_{2}$)	$\alpha_{2}=6\left(\frac{\varepsilon \ell_{b}^{4}}{Eh^{3}d_{l}^{3}}\right)$

**Table 3 micromachines-14-00903-t003:** Mid-point rise for each design in µm.

	Design 1	Design 2	Design 3	Design 4	Design 5
$b_{l\circ }$	0	0	2	−2	−2.5
$b_{u\circ }$	1	2	2	0	2

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
