# Peer review of "Static and Eigenvalue Analysis of Electrostatically Coupled and Tunable Shallow Micro-Arches for Sensing-Based Applications"

_micromachines, 2023, doi:10.3390/mi14050903_

Round 1

Reviewer 1 Report

I think that the authors present a solid model and some of the results are rather interesting. However, the manuscript presentation is horrible and an overhaul revision is needed. The following is some presentation issues. To be honest, I did not finish the reading.

1.       In the governing equations of Eqs. (1) and (3), there are +- signs. The authors should explain when + and – should be used in conjunction with the two beam configurations as presented in Fig. 1. Otherwise, quite some readers will be confused. Furthermore, the authors should explain the physical meaning of the variables as presented in these two equations, which makes things easier to be understood. It seems to me that the variables used in these equations are NOT consistent with those presented in Table 1. For example, the initial imperfections. In Table 1, it should be called the initial imperfections not initial curvature because curvature does not have the length unit of micron.

2.       Just below Eq. (1), the bracket [] rather than double parentheses (()) should be used. Furthermore, the initial imperfections are assumed to be the shape of the first mode shape. I am just wondering what are the impacts of the other shapes? Should the coupling and mode localization be enhanced?

3.       In Table 2, “Viscus should be Viscous.

4.       Line 107, “where A is the beam element cross-sectional area. As presented in Table 1, we can see that the beam is a uniform one. Here the word element can cause nothing but confusion. You can just say that “ where A is the cross-section area of the beam.”

5.       Line 119, the sentence “A reduced-Order model (ROM) based on a Galerkin approximation. Firstly, ““A reduced-Order model” should be ““A reduced-order model”. Secondly, there is no so-called “a Galerkin approximation”, there is only the Galerkin method. The approximation/error is due to the truncation error in the Galerkin method.

6.       What is udc in Figure 2? In the text (line 180) you use udc. And what is ve w_s in Fig. 2? In Fig.2, the dimensional results are presented. If you want to present the dimensional results, why bother to do the lengthy non-dimensionalization procedures from Eq. (5) to Eq. (17).  

Author Response

Dear Respected Reviewer,

Thank you very much for your helpful comments. These professional comments are indispensable for improving the quality of the paper. This paper has been carefully revised. And all the changes are highlighted in yellow in the revised manuscript. The following are our responses to the report comments.

Sincerely,

The authors

4/13/2023

Reviewer 2 Report

This paper investigated the mechanical performance of an electrostatically coupled and tunable microbeams-based resonator, which have potential for use in a wide range of MEMS applications. The manuscript is interesting and can be recommended for publication after following minor revisions.

1. The novelty of this work should be emphasized more. The author may explain deficiencies or shortcomings of other studies to make a bridge to introducing the novelty of their work, especially?

2. There are too many paragraphs in Introduction, and the Introduction did not give a clear review on the topic. It is suggested the Introduction should be further revised in terms of contents and logic

3. How to verify the accuracy and correctness of the obtained result?

4. The fundamental frequency of the system is studied instead of the resonance behavior. The statement on resonance frequency is suggested to revised.

5. The following studies on this topic may be helpful to enhance the literature review quality of this paper:

10.1016/j.ijnonlinmec.2023.104369;

10.1016/j.compstruct.2023.116709;

10.1007/s10483-022-2917-7

Author Response

(The authors gave the same response as above.)

Reviewer 3 Report

The authors investigated the mechanical performance of two electrostatically coupled micro-arches. The paper is interesting with a potential implication for mode-localized MEMS sensors. The authors must perform some modifications by addressing the following comments:

1. In the introduction, the authors must enrich the literature survey with several papers dealing with electrostatically coupled MEMS resonators, which is a topic deeply investigated in the recent past. 

2. In the equations, the authors did not include the fringing field effect. If it is neglected, the authors must justify it.

3.  For the reduced order model, the authors did not justify the selected modes in the Galerkin procedure. How the authors know that convergence is reached?

4.  The authors can add few lines about the stability analysis.

5. What about the sensitivity of the proposed design to temperature fluctuations?

6. The proposed design could be very sensitive in some operating points, which makes its robustness questionable. The authors must discuss this issue.

7. The presented study is limited to the effect of DC voltage. It would be interesting if the authors complete it with some figures showing the effects of the initial beam curvatures.

8. The authors should reduce the number of self-citations

Author Response

Dear Respected Reviewer,

Thank you very much for your helpful comments. These professional comments are indispensable for improving the quality of the paper. This paper has been carefully revised. And all the changes are highlighted in yellow in the revised manuscript. The following are our responses to the report’s comments.

Sincerely,

The authors

4/13/2023

Round 2

Reviewer 1 Report

The authors addressed my comments one by one. Overally I am satisfied with their efforts/ 

Author Response

Dear Respected Reviewer, 

Thank you for your kind words.

The authors,

Reviewer 3 Report

The authors did not carefully address my first comment. The state of the art is rich with papers studying electrostatically coupled resonators. The authors must acknowledge the recent literature related to the highlight of the paper such as: 

(i) https://doi.org/10.1109/ICSENS.2016.7808509

(ii) https://doi.org/10.1016/j.ijnonlinmec.2019.103366

(iii) https://doi.org/10.1016/j.ymssp.2021.107781

Author Response

Dear Respected Reviewer, 

We would like to thank you for your valuable comments. The revised version has addressed your comment. The introduction has been updated to cover various related works in MEMS-coupled resonators.

The changes are highlighted in the submitted manuscript. 

Sincerely, 

The authors